# Exploring the hydrophobic effects of quaternary ammonium copolymers on corrosion of casing and tubing steel in acidic solution

Ghadeer Mubarak[1], Chandrabhan Verma [1]*, Mohammad A. Jafar Mazumder[2,3], Imad Barsoum[4], Akram Alfantazi[1]*

1 Department of Petroleum and Chemical Engineering, Khalifa University of Science and Technology, Abu Dhabi, United Arab Emirates, 2 Department of Chemistry, King Fahd University of Petroleum & Minerals, Dhahran, Saudia Arabia, 3 Interdisciplinary Research Center for Refining and Advanced Chemicals, King Fahd University of Petroleum & Minerals, Dhahran, Saudi Arabia, 4 Department of Mechanical Engineering, Khalifa University of Science and Technology, Abu Dhabi, United Arab Emirates

* chandraverma.rs.apc@itbhu.ac.in (CV); akram.alfantazi@ku.ac.ae (AA)

## Abstract

By encouraging improved adsorption onto metal surfaces and creating a more powerful barrier against corrosive chemicals, the hydrophobic property of corrosion inhibitors raises inhibition efficiency and decreases corrosion rates. This work aims to synthesize and describe three quaternary ammonium-based copolymers (AMCs) with different hydrophobic qualities and investigate their ability to inhibit P110 CS corrosion in 15% HCl, which is helpful for casing and tubing. The results showed that AMCs act as efficient corrosion inhibitors, with over 90% inhibition efficiency (%$IE$) at 20 ppm concentration. The electrochemical investigation results indicated that the AMCs with hydrophilic and hydrophobic ratios of 100 (**5**), 90:10 (**6a**), and 80:20 (**6b**) manifest %$IE$ of 87.74%, 92.12%, and 93.53%, respectively. The electrochemical investigations show that at the metallic surface's active areas, AMCs successfully replace the pre-adsorbed water molecules. They are categorized as mixed-type corrosion inhibitors because they prevent both anodic and cathodic reactions without appreciably changing the corrosion potential ($E_{corr}$). Their adsorption on the metallic surface follows the Langmuir adsorption isotherm. Surface analysis tools like SEM and EDX are utilized to investigate the corrosion prevention mechanism of adsorption. The DFT analysis results show that quaternary nitrogen atoms of hydrophilic and hydrophobic moieties play a key role in the adsorption and charge-sharing processes. Finally, the corrosion prevention mechanism of AMCs is explained using a graphic depiction based on the ideas of electrochemical, surface, and computational studies.

**Data availability statement:** All relevant data are within the manuscript and its Supporting Information files.

**Funding:** The author(s) received no specific funding for this work.

**Competing interests:** The authors have declared that no competing interests exist.

## 1. Introduction

The oil and gas industries have significant challenges due to casing and tubing corrosion, which arises from their exposure to harsh and caustic environments during exploration and production [1,2]. Increased corrosion rates brought on by a combination of corrosive fluids, high temperatures, and mechanical stress could compromise the integrity of the well-bore infrastructure. Not only does this pose a structural concern, but hazardous compounds have the potential to leak out of the casing and tubing, causing environmental risk. Since downhole conditions are complex and applying effective corrosion prevention methods is challenging, special materials, corrosion inhibitors and continuous monitoring are needed to minimize damage and maintain the long-term reliability of oil and gas wells [3]. Carbon dioxide ($CO_2$) and hydrogen sulfide ($H_2S$) in production fluids are the main causes of casing and tubing corrosion in oil and gas wells [4–6]. Substances like hydrogen sulfide, organic acids (OAs), dissolved carbon dioxide, and other acidic compounds commonly found in hydrocarbon reservoirs can create acidic conditions, with pH levels ranging from mildly to strongly acidic. While $CO_2$ and OAs contribute to corrosion by lowering the pH, hydrogen sulfide is particularly aggressive and can lead to sulfide stress corrosion cracking [4–6]. Factors such as bacteria, oxygen, and chloride ions in the production fluids can further aggravate corrosion. Therefore, developing corrosion inhibitors specifically designed for 15% hydrochloric acid (HCl) is critical in the oil and gas industry [7]. In well stimulation, 15% HCl is often used for acidizing because it dissolves blockages in the formation, increasing hydrocarbon production. However, HCl is highly corrosive and can seriously damage well components like casings and tubing. To protect these parts and ensure the long-term safety and reliability of oil and gas infrastructure, effective corrosion inhibitors for 15% HCl are essential [8,9]. These inhibitors help reduce metal damage, minimize downtime, maintain operational efficiency and lower maintenance costs during acidizing.

P110 carbon steel is commonly used for these components due to its strength and resistance to high pressure, but it is still prone to corrosion in harsh oilfield environments. P110 steel is generally preferred over the other steel grades because of its balanced combination of toughness, cost-effectivity and mechanical strength. Compared to other grade steels, including N80 or J55, P110 CS is associated with high yield and tensile strength, which is essential for the cost-effective design of thin-walled steel structures for casing and tubing applications. P110 CS is also associated with high resistance to deformation under high stress and mechanical wear. Several methods help prevent corrosion in P110 steel. One common approach is using corrosion inhibitors, like organic compounds, that form a protective layer on the steel. Another approach is applying coatings, such as epoxy or polymer, to act as a barrier against corrosive substances. Cathodic protection methods, such as sacrificial anodes and impressed current systems, also help by altering the steel's electrochemical environment to slow corrosion [10,11]. However, using organic corrosion inhibitors presents challenges due to reservoir conditions like composition, pressure, and temperature that can affect their performance [3,5]. It's also difficult to ensure even distribution of the inhibitor throughout the entire length of tubing and casing, especially in complex

well designs. Additionally, inhibitors must be stable and resistant to breakdown for long-term protection in harsh environments [3,5]. Issues like compatibility with other well fluids and the risk of inhibitors settling or sticking to solids in the reservoir further complicate their effective use [3,5]. Surfactants have gained attention as effective corrosion inhibitors with their unique molecular structure consisting of hydrophobic and hydrophilic parts [3,5]. Their amphiphilic nature allows them to stick to metal surfaces and form a protective layer that reduces corrosion. Recent studies have focused on improving surfactant formulations to enhance their protective abilities and tailor them for specific environments and metal types. Researchers have also explored combining surfactants with other inhibitors or nanoparticles to boost performance.

Due to their versatility, ability to lower surface tension, and advances in understanding through computer modeling, surfactants are increasingly important in corrosion prevention across various industries. In the oil and gas sector, surfactants have shown great potential in protecting casing and tubing from corroding. They attach to metal surfaces, forming a protective layer that blocks corrosive elements. Recent research [12–14] has focused on developing surfactants to withstand harsh oilfield conditions, such as high pressures and temperatures. Their good wetting properties ensure even coverage across tubing and casing surfaces [12–14]. They also offer an affordable, eco-friendly solution that can help extend the lifespan and integrity of casing and tubing systems in oil and gas wells [15].

The literature study indicates that corrosion inhibitors' hydrophobic or hydrophilic properties play crucial roles in determining their solubility and efficiency. A suitable combination of hydrophilic and hydrophobic segments is essential for desirable corrosion protection. In the present study, three surfactant molecules with different hydrophilic and hydrophobic properties were synthesized, characterized and tested as corrosion inhibitors for mild steel in a 15% HCl solution due to their high solubility, strong adsorption and effective corrosion protection. The study aimed to show how increasing the hydrophobic segments of quaternary ammonium copolymers (AMCs) affects their ability to prevent corrosion of P110 carbon steel in 15% HCl. This highlights the need to carefully balance a molecule's hydrophilic and hydrophobic parts before using it as a corrosion inhibitor. The findings provide insights for the industry to identify and formulate corrosion inhibitors based on surfactants most suited for 15% HCl conditions.

## 2. Experimental testing

### 2.1. Polymer synthesis

The following materials were purchased and used as received: Diallyl amine, N,N-dimethyldodecylamine, 1,6-dibromohexane, and ammonium persulfate (APS) from Sigma-Aldrich; Acetonitrile, diethyl ether, acetone, 37% concentrated HCl, and sodium hydroxide from Fluka Chemie AG. Monomer 4 was synthesized using the previously reported method [16], and the polymer was purified through dialysis using Spectra/Por membrane (MWCO: 3.5 kDa) from Spectrum Laboratories Inc.

### 2.2. Characterization techniques

The structures of the synthesized compounds were characterized using a PerkinElmer Series II Model 2400 FTIR spectrometer, with 32 scans and a spectral resolution of $4\ cm^{-1}$. The $^{1}$H and $^{13}$C NMR spectra were recorded on a Bruker Avance III 400 MHz spectrometer. Melting points were determined using an Electrothermal-SMP30 digital melting point apparatus at a rate of $1\ °\frac{C}{min}$ hermogravimetric analysis (TGA) was performed under nitrogen using a SDT analyzer (Q600: TA instruments) from 20 °C to 800 °C.

**2.1.1. *Synthesis of quaternary ammonium bromide (3)*.** Quaternary ammonium bromide (3) was synthesized based on a modified method from the literature [17]. N,N-dimethyldodecylamine 1 (10.7 g, 50 mmol), and 1,6-dibromohexane 2 (48.8 g, 200 mmol) were heated at 65 °C in acetonitrile (80 mL) under N 2 for 24 h. After removal of the excess dibromohexane, the remaining mixture was washed with diethyl ether, dissolved in hot acetone, and cooled to participate in the bis-quaternary salt. The mono-carionic salt (3) was obtained as a white solid (17.2 g, 75%) with a melting point of 58–59 °C. FTIR and NMR spectra confirmed its structure in Figures SI1 and SI 2 [S1 File].

**2.1.2. *Synthesis of diallylammonium chloride and cyclopolymer.*** Cyclopolymer 5 was synthesized following the procedure published in the earlier literature [18]. Diallyamine (10.02 g, 103 mmol) was cooled to 0 °C, and 37% concentrated HCl (11.8 g, 120 mmol) was added slowly to form a solution of diallyammonium chloride (4). The mixture was heated to 100 °C under nitrogen, and APS (400 mg) was added in four potions to initiate the polymerization. After 20 minutes of heating at 105 °C, the solution was dialyzed for 18 hours and freeze-dried to yield the cyclopolymer 5 as a white solid (11.6 g, 84.7%). FTIR and NMR confirmed the polymer structure in Figures SI3-S5 [S1 File].

**2.1.3. *Synthesis of 6a.*** Methanol (12 mL), cyclopolymer 5 (4.00 g, 30 mmol), and NaOH (0.20 g, 5 mmol) were stirred to form a white paste. Acetonitrile and toluene were added, followed by quaternary ammonium bromide (3). The mixture was heated at 70 °C for 24 hours under nitrogen. After solvent removal and dialysis, the hydrophobically modified copolymer 6a was obtained as a creamy solid (4.50 g, 91%). FTIR and NMR confirmed the structure in Figure SI6 [S1 File].

**2.1.4. *Synthesis of 6b.*** The synthesis of 6b followed the same procedure as 6a, using methanol (8 mL), cyclopolymer 5 (2.67 g, 20 mmol), quaternary ammonium bromide (3), NaOH, acetonitrile, and toluene. This yielded the copolymer 6b as solid (3.60 g, 90%). The FTIR spectrum and NMR signals confirmed the structure in Figures SI7, SI8 and SI9 [S1 File].

## 2.2. *Material, electrolyte and inhibitor concentrations*

For electrochemical (OCP, EIS, and PDP) analyses, industrial-grade P110 carbon steel (P110 CS) was utilized as the working material. Before the experiments, emery (Si-C) papers with grid sizes ranging from 600 to 1200 were used to polish the P110 CS surfaces. After de-greasing with acetone and drying under a hot-air blower, the surface was polished, cleaned with distilled water, and then preserved in moisture-free desiccants. The test electrolyte was a 15% hydrochloric acid solution. Following tests at concentrations ranging from 5–20 ppm, the corrosion inhibition potential of AMCs was optimal at 20 ppm. The evaluated inhibitors were completely soluble in the test electrolyte within the investigated concentration range because of their cationic nature. The data exhibiting the most consistent patterns were utilized in the investigation.

## 2.3 Electrochemical, surface and DFT studies

The Gamry Potentiostat/Galvanostat (Model G-300) with EIS software from Gamry Instruments Inc., USA, and the Echem Analyst 5.0 software package were used for all open circuit potential (OCP), potentiodynamic polarization (PDP) and electrochemical impedance spectroscope (EIS) measurements. A typical three-electrode glass cell with a P110 CS specimen with a $1\ cm^2$ (one-sided) exposed surface was used as the reference electrode (RE), graphite rode counter electrode (CE), and working electrode (WE) for all electrochemical testing. After stabilizing OCP with a 30-minute immersion period, the OCP, EIS and PDP studies were carried out. AC signals with a peak-to-peak amplitude of 10 mV were used to record EIS spectra at open circuit potential in the $100\ kHz\ to\ 0.01\ Hz$ frequency range. Equation (1) was utilized to obtain the respective inhibition efficiencies ($\%IE_{EIS}$) after precise fitting and simulation studies of the spectra were performed. A current of $\pm 250\ mV$ about OCP and a sweep rate of $1.0\ mV\ s^{-1}$ were employed in the polarization investigation. The instrument's installed Echem Analyst 5.0 software application was used to extrapolate polarization parameters, such as anodic ($\beta_a$) and cathodic ($\beta_c$) Tafel slopes, corrosion potential ($E_{corr}$), and corrosion current ($i_{corr}$), using the method of choice. The following relationship was applied to the $i_{corr}$ readings to determine the $\%IE_{PDP}$, and the appropriate surface coverage values were computed. [19,20]:

$$\%IE_{PDP} = \frac{i^0_{corr} - i^i_{corr}}{i^0_{corr}} \times 100$$

(1)

$$\%IE_{EIS} = \frac{R^i_{ct} - R^0_{ct}}{R^i_{ct}} \times 100$$

(2)

where, $i_{corr}$ and $R_{ct}$ represent the carrion current density and charge transfer resistance, respectively. "0" denotes the lack of inhibitors while "i" denotes their presence.

To characterize the adsorption manner of corrosion inhibition by DAC-DAD10 and DAC-DAD20, surface investigations utilizing SEM (scanning electron microscope) combined with EDX (energy dispersive X-ray) were conducted. Using a FEI Nova NanoSEM 650 Scanning Electron Microscope that was operating in high vacuum and fitted with a NiCol electron column and Trinity detector, the morphology of P110 CS samples was examined. The beam included a resolution of 1.0 nm, a working distance of 10 mm, a beam current of 10 pA, and an acceleration of 15 kV. The elemental contents of the samples were also measured using SEM-EDX. Gaussian 09 software was used to do the DFT study. The various parameters were derived using the following relationships [21,22]:

$$E_{HOMO} = -IP \tag{3}$$

$$E_{LUMO} = -EA \tag{4}$$

$$\Delta E = E_{LUMO} - E_{HOMO} \tag{5}$$

$$\chi = \frac{IP + EA}{2} \tag{6}$$

$$\eta = \frac{\Delta E}{2} \tag{7}$$

$$\sigma = \frac{1}{\eta} \tag{8}$$

$$\Delta N = \frac{\Phi_{Fe} - \chi_{inh}}{2\left(\eta_{inh} + \eta_{Fe}\right)} \tag{9}$$

In the above equations, $E_{HOMO}$, $E_{LUMO}$, $IP$, $EA$, $\Delta E$, $\chi$, $\eta$, $\sigma$ and $\Delta N$ represent the energy of HOMO (highest occupied molecular orbital), energy of LUMO (lowest unoccupied molecular orbital), ionization potential, electron affinity, electronegativity, hardness, softness and fraction of electron transfer, respectively. For the calculation of $\Delta N$, $\Phi_{Fe}$ of $4.82\ eV$ and iron's global hardness value of zero were considered [21–23].

## 3. Results and discussions

### 3.1. Polymer synthesis and characterization

To create the cationic salt bromide 3, excess dibromohexane 2 was combined with *N, N-dimethyl* dodecyl amine 1. To produce excellent yields of polymer 5, diallyl amine salt 4 underwent cyclopolymerization. As shown in Scheme 1, polymer 5 was modified hydrophobic/hydrophilic to produce copolymers 6a and 6b. Due to the complicated ionic nature of the polymer, the molecular weight determination of the polymer by GPC was unsuccessful. The Ubbelohde viscometer determined the viscometric behavior of the polymers in $0.1\ M\ NaCl\ at\ 30 \pm 0.1\ °C$. The intrinsic viscosity [η] values were determined to be $0.105,\ 0.121$ and $0.080\ dL\ g^{-1}$ for homopolymer 5 and copolymers 6a (10% incorporation), and 6b (20% incorporation),

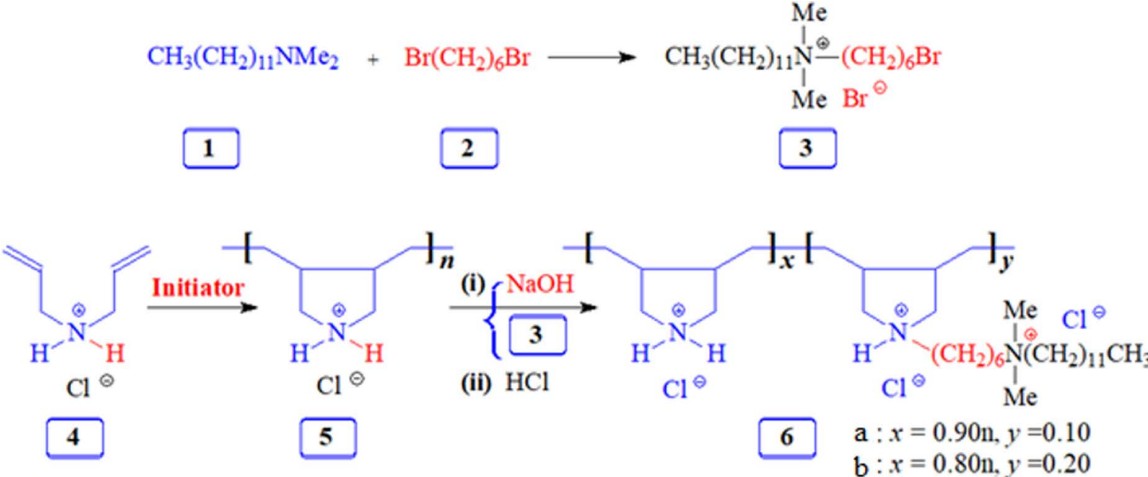

**Scheme 1. Synthesis of polymers 5 and 6.**

respectively. An excessive number of hydrophobic pendants presumably leads to self-micellization, decreasing its hydrodynamic volume and viscosity. An increase in hydrophobe content enhances intramolecular association [24,25]. In the lower hydrophobe content range, the distance between the hydrophobe units in the polymer backbone increases, thereby preventing them from undergoing intramolecular association. Recall that the alteration resulted in pendants with hydrophilic quaternary ammonium and hydrophobic alkyl chain in number six, which are predicted to demonstrate corrosion inhibition properties. The proton, carbon count, and area integration of the proton signals (supplementary information) confirmed the synthesis of the polymer structure. Besides, the elemental analysis of quaternary ammonium salt **3** closely matches the calculated values, ensuring the synthesis of the structure. The TGA (Figure SI10 [S1 File]) indicated that the homo and copolymers were thermally stable up to 250 °C, encouraging high temperatures in corrosion applications.

### 3.2. Electrochemical studies

**3.2.1. *Open circuit potential (OCP) study*.** Open circuit potential refers to the voltage of a corroding metal or alloy when it is not connected to any external circuit [26]. Fig 1 shows the OCP vs. time curves for P110 CS corrosion in a 15% HCl solution with and without AMCs. As previously mentioned, the OCP vs time curve measurements were made following a 30-minute electrode immersion. After closely examining the OCP vs. time curves, it is clear that the OCP vs. time curves are strong throughout the measurement. Generally, this type of observation is attributed to the replacement of pre-adsorbed water molecules by the inhibitor molecules [27,28]. Additionally, a notable shift in OCP about time curves is evident when OCP is present. However, the shift was inconsistent; the OCP of the blank shifted towards a negative direction at some concentrations and in a positive direction at others.

**3.2.2. *Potentiodynamic polarization (PDP) study*.** Potentiodynamic polarization studies can provide an in-depth understanding of the electrochemical behavior of metallic materials in corrosive environments [29]. This experimental method measures the current that results from consistently altering the electrode potential to generate polarization curves. In 15% HCl, potentiodynamic polarization experiments were carried out for P110 CS corrosion with and without AMCs at varying concentrations. Table 1 represents the polarization parameters. Fig 2 displays the anodic and cathodic polarization (Tafel) curves. The polarization curves exhibited a notable alteration or shift compared to the blank. The results presented in Table 1 showed that the presence of AMCs significantly affects the corrosion current density ($i_{corr}$). According to the noted significant shift in $i_{corr}$, adding AMCs significantly impacts how P110 CS behaves when it corrodes

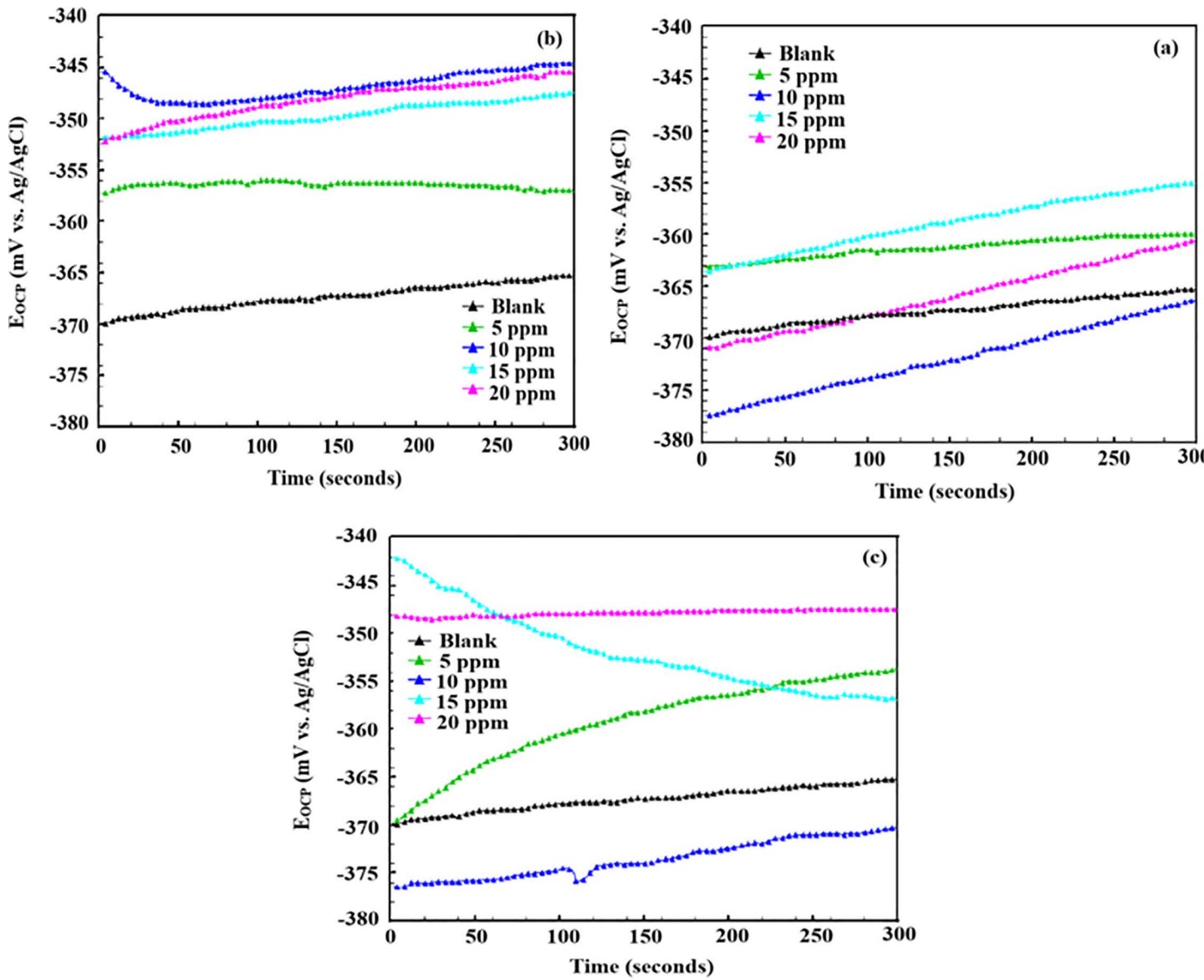

**Fig 1. OCP vs. time curves for P110 carbon steel corrosion in the absence and presence of different concentrations of** (a) 5, (b) 6a and (c) 6b.

in 15% HCl. The substantial decrease in corrosion observed when inhibitors are present suggests that AMCs slow down corrosion by attaching to the active sites on the metallic surface [30,31]. Further, examination reveals that inhibitors have no discernible effect on the corrosion potential ($E_{corr}$). Corrosion inhibitors are frequently categorized according to how they affect the corrosion potential of the corroding system. They are characterized as anodic or cathodic-type if the shift in $E_{corr}$ in the presence of inhibitors is more significant than $85\ mV$. However, they are categorized as mixed-type if this shift is less than $\pm 85\ mV$. The analysis of the study's results shows that the corrosion potential values shift by less han $\pm 85\ mV$, which suggests that the studied inhibitors can be categorized as mixed-type [32,33].

### 3.2.3. *Electrochemical impedance spectroscope (EIS) study.* Electrochemical impedance spectroscopy (EIS)
is a powerful technique for a highly precise analysis of metal corrosion in various electrolytes [34,35]. In an EIS study,

**Table 1. Potentiodynamic polarization parameters for P110 carbon steel corrosion in 15% HCl with and without different concentrations of AMCs at 298K.**

| Inhibitor | Conc. (ppm) | $E_{corr}$ (mV) | $i_{corr}$ (mA/cm²) | $\beta_a$ $\frac{V}{decade}$ | $-\beta_c$ $\frac{V}{decade}$ | Corrosion rate $C_R$ mpy | %$IE_{PDP}$ | Surface cover (θ) | Chi-Squared |
|---|---|---|---|---|---|---|---|---|---|
| Blank | -- | −363.0 | 1300 | 81.70e-3 | 226.1e-3 | 592.9 | -- | -- | 34.21 |
| 5 | 5 | −371.0 | 417.0 | 188.7e-3 | 663.8e-3 | 190.5 | 67.86 | 0.6786 | 8.096 |
|  | 10 | −356.0 | 285.0 | 73.20e-3 | 171.0e-3 | 130.1 | 78.05 | 0.7805 | 1.766 |
|  | 15 | −345.0 | 159.0 | 65.00e-3 | 137.3e-3 | 72.68 | 87.74 | 0.8774 | 5.507 |
|  | 20 | −353.0 | 133.0 | 64.70e-3 | 116.7e-3 | 60.66 | 89.71 | 0.8971 | 6.729 |
| 6a | 5 | −361.0 | 320.0 | 157.2e-3 | 258.5e-3 | 146.3 | 75.32 | 0.7532 | 4.114 |
|  | 10 | −359.0 | 180.0 | 86.40e-3 | 229.7e-3 | 82.27 | 86.12 | 0.8612 | 1.152 |
|  | 15 | −342.0 | 143.0 | 75.20e-3 | 169.6e-3 | 65.47 | 88.95 | 0.8895 | 7.372 |
|  | 20 | −348.0 | 102.0 | 77.50e-3 | 151.8e-3 | 46.68 | 92.12 | 0.9212 | 6.840 |
| 6b | 5 | −354.0 | 288.0 | 96.80e-3 | 206.3e-3 | 131.5 | 77.82 | 0.7782 | 3.239 |
|  | 10 | −351.0 | 168.0 | 69.60e-3 | 203.2e-3 | 76.57 | 87.08 | 0.8708 | 5.116 |
|  | 15 | −339.0 | 134.0 | 67.50e-3 | 121.2e-3 | 61.23 | 89.68 | 0.8968 | 18.15 |
|  | 20 | −355.0 | 83.90 | 51.20e-3 | 74.10e-3 | 38.33 | 93.53 | 0.9353 | 34.27 |

an electrolyte-immersed, corroding metal electrode is subjected to many frequencies of small amplitude alternating current. Examining the resulting impedance response provides crucial information into the corrosion processes at the metal/solution interface [34,35]. Typically, the EIS spectrum consists of two primary components: the inductive behavior connected to the diffusion of corrosion products and the capacitive behavior related to the metal/electrolyte interface. Fig 3 and 4 show the Nyquist and Bode (frequency and phase angle) diagrams with and without AMCs at various concentrations. The Nyquist curves show that P110 CS corrosion in 15% HCl involves a single charge transfer process since they offer a single semicircle in the inhibitor molecule-free and inhibitor-containing conditions. The Bode phase angle plots of P110 CS corrosion in 15% HCl solution likewise show single maxima, indicative of a single charge transfer mechanism. Important information about the dynamic behavior of the corrosion processes can be gleaned from the Bode frequency curves with and without corrosion inhibitors.

Corrosion inhibitors affect Bode plots, particularly in the low-frequency range, because they change the corrosion mechanisms and lead to the growth of protective layers on the metal surface. In the absence of AMCs, the phase angle value is smaller than when they are present, according to a detailed analysis of the Bode phase angle graphs. A smaller magnitude of phase angle value is usually associated with a rougher and highly corroded surface. Without AMC molecules, the P110 CS surface is observably more corroded and damaged due to attacks by free acids [36,37]. Several helpful characteristics, such as solution resistance ($R_s$), charge transfer resistance ($R_{ct}$), phase shift ($n$), and a constant phase element (CPE), were obtained by fitting the Nyquist curves to an equivalent circuit described in our previous study [18]. For acid-based electrochemical corrosion of metals, CPE frequently produces better approximations. The CPE's impedance ($Z_{CPE}$) and double layer capacitance ($C_{dl}$) values can be shown in the following way [36,37]:

$$Z_{CPE} = \left(\frac{1}{Y_0}\right)\left[(j\omega)^n\right]^{-1} \tag{10}$$

$$C_{dl} = Y_0[(\omega_{max})]^{n-1} \tag{11}$$

where, $n$ is the phase shift (exponent), $j$ is the imaginary number ($j^2 = -1$), $\omega$ is the angular frequency, and $Y_0$ is the CPE constant. The findings indicate that the presence of AMC molecules significantly raises the values of charge resistance,

 

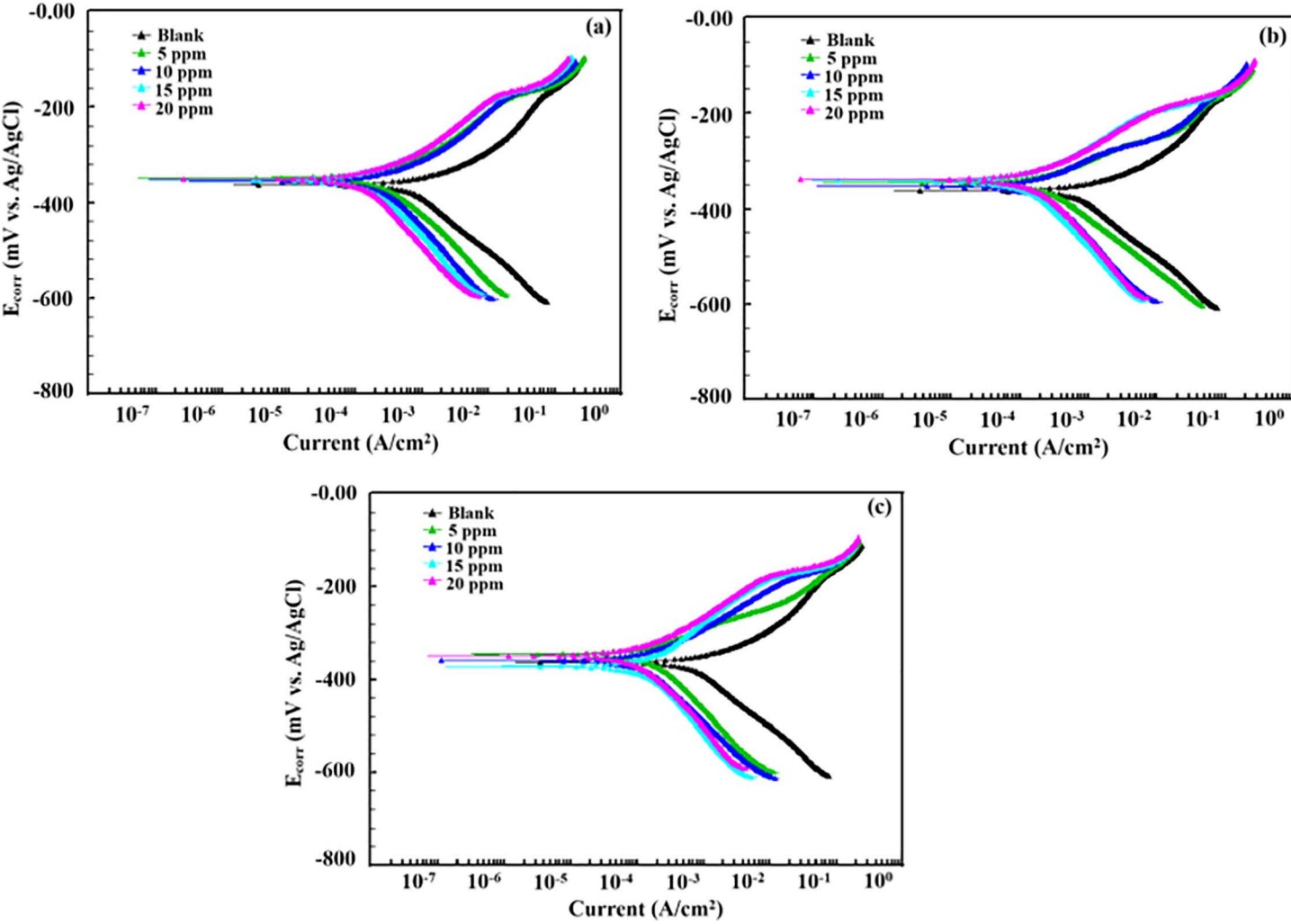

**Fig 2. Potentiodynamic polarization curves for P110 carbon steel corrosion in the absence and presence of different concentrations of** (a) 5, (b) 6a and (c) 6b.

which measures the impedance to electron transfer at the metal-electrolyte interface. An enhanced $R_{ct}$ is frequently linked to a disruption in the corrosion reactions. The higher $R_{ct}$ suggests that the inhibitor has successfully slowed down the mechanisms involved in charge transfer [38]. This is attribute to the formation of a protective film or a decrease in the rate of corrosive reactions at the metal surface. A greater $R_{ct}$, which denotes a more substantial barrier against corrosion, indicates the inhibitor's efficiency. The examination of results presented in Table 2 reveals that the presence of AMCs has significantly decreased in $C_{dl}$ values. The drop in $C_{dl}$ indicates a change in the electrochemical processes since it suggests that the metal surface's capacity to store charge has diminished [39,40]. This implies that the inhibitor blocks corrosive species from accessing the metal by creating a denser, more protective barrier on the metal surface. The stabilization of the metal-electrolyte interface by the inhibitor is commonly linked to enhanced inhibition efficiency [41].

### 3.3. *Thermodynamic and kinetic studies*

The extent of interaction between inhibitor (AMC) molecules and the metal surface can be explained by adsorption isotherms [42]. The formation of protective layers on metal surfaces is attributed to the adsorption of inhibitor molecules.

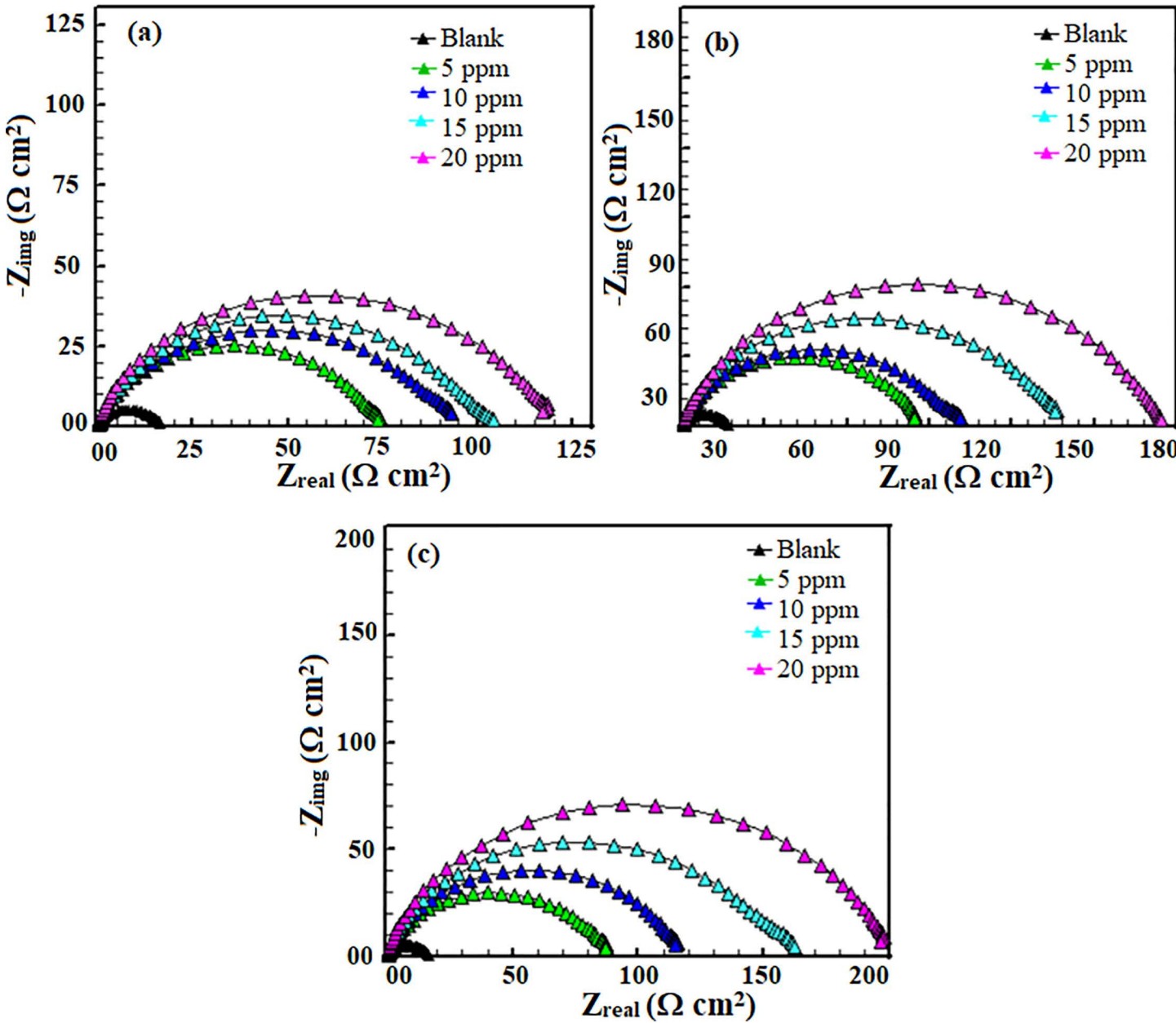

**Fig 3. Nyquist plots for P110 carbon steel corrosion in the absence and presence of different concentrations of** (a) 5, (b) 6a and (c) 6b.

The actual adsorption of the inhibition mechanism can be verified by applying isotherm equations. In addition to being straightforward, the Temkin, Freundlich and Langmuir isotherms can easily obtain comprehensive information from their parameters to describe the corrosion inhibition system. In the present study, the adsorption behavior of AMCs on the P110 CS surface in the 15% HCl solution was tested using these isotherm models. The fitting parameters for these isotherm models, including intercept, slope and regression coefficient ($R^2$) values, are presented in Table 3 and different tested isotherm models are shown in Fig 5a-c. To determine the most suitable isotherm model for the adsorption of AMCs on the

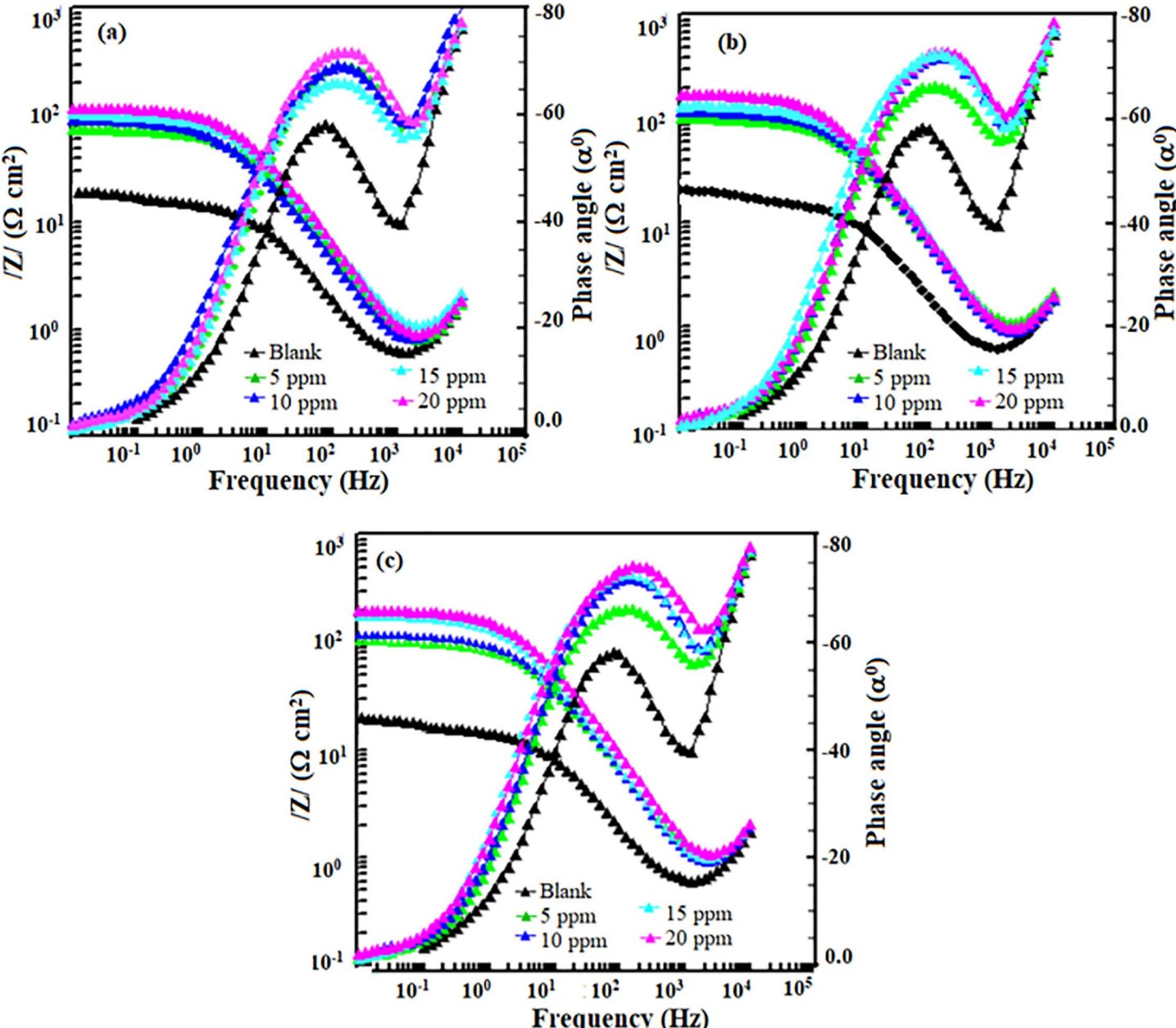

**Fig 4. Nyquist plots for P110 carbon steel corrosion in the absence and presence of different concentrations of** (a) 5, (b) 6a and (c) 6b.

P110 CS surface in a 15% HCl solution, the $R^2$ values were employed. As the values of $R^2$ for this model were closest to unity, the results demonstrate that the Langmuir isotherm showed the best fit for all investigated inhibitor compounds.

The Langmuir, Temkin and Freundlich isotherms equation can be presented as follows [20,43]:

$$K_{ads}C = \frac{\theta}{1-\theta}$$

$$(12)$$

**Table 2. Electrochemical impedance spectroscope parameters for P110 carbon steel corrosion in 15% HCl with and without different concentrations of AMCs at 298K.**

| Inhibitor | Conc. (ppm) | $R_s$ ($\Omega$/cm²) | $R_{ct}$ ($\Omega$/cm²) | $Y_0$ | $\%IE_{EIS}$ | $\theta$ | $n$ | $C_{dl}$ ($\mu$F/ cm²) | Goodness of fit |
|-----------|-------------|----------|-----------|-------|--------------|----------|-----|-------------|-----------------|
| Blank | -- | 0.311 | 15.749 | 691.7 | -- | -- | 0.758 | 157.437 | 5.894e-3 |
| 5 | 5 | 1.87 | 72.49 | 216.9 | 78.27 | 0.7827 | 0.763 | 48.941 | 55.66e-3 |
| | 10 | 1.28 | 91.42 | 315.4 | 82.77 | 0.8277 | 0.774 | 77.095 | 71.01e-3 |
| | 15 | 2.04 | 101.36 | 224.7 | 84.46 | 0.8446 | 0.736 | 46.936 | 55.48e-3 |
| | 20 | 1.94 | 116.36 | 154.6 | 86.46 | 0.8646 | 0.782 | 40.570 | 56.73e-3 |
| 6a | 5 | 2.31 | 85.46 | 183.3 | 81.57 | 0.8157 | 0.771 | 74.263 | 54.39e-3 |
| | 10 | 2.23 | 99.23 | 169.5 | 84.12 | 0.8412 | 0.764 | 41.915 | 55.52e-3 |
| | 15 | 1.73 | 135.6 | 132.9 | 88.38 | 0.8838 | 0.785 | 34.413 | 56.74e-3 |
| | 20 | 2.16 | 175.1 | 141.1 | 91.00 | 0.9100 | 0.765 | 32.489 | 55.78e-3 |
| 6b | 5 | 2.55 | 87.76 | 178.2 | 82.05 | 0.8205 | 0.754 | 38.704 | 50.52e-3 |
| | 10 | 2.56 | 115.9 | 131.4 | 86.41 | 0.8641 | 0.781 | 34.598 | 49.14e-3 |
| | 15 | 1.73 | 157.5 | 122.8 | 90.00 | 0.9000 | 0.779 | 31.665 | 47.57e-3 |
| | 20 | 1.83 | 199 | 99.93 | 92.08 | 0.9208 | 0.794 | 26.574 | 51.46e-3 |

**Table 3. The fitting parameters (intercept, slope and regression coefficient) of different tested isotherm models for the adsorption of AMCs on P110 CS surface in 15% HCl.**

| Inhibitor | Isotherm model | | | | | | | | |
|-----------|-----------------------------|--------|--------|-------------------------|--------|--------|-------------------------|--------|--------|
| | Freundlich Isotherm | | | Temkin Isotherm | | | Langmuir Isotherm | | |
| | Intercept | Slope | $R^2$ | Intercept | Slope | $R^2$ | Intercept | Slope | $R^2$ |
| 5 | 0.0045x | −.5399 | 0.9206 | 0.0150x | 0.6203 | 0.9356 | 0.9812x | 2.6272 | 0.9981 |
| 6a | 0.0029x | −.5063 | 0.8661 | 0.0106x | 0.7232 | 0.8872 | 0.9897x | 1.7524 | 0.9992 |
| 6b | 0.0027x | − 0.499 | 0.9043 | 0.0099x | 0.7459 | 0.9221 | 1.0024x | 1.4752 | 0.9994 |

$$K_{ads}C_{inh} = \theta \tag{13}$$

$$\theta = -\frac{1}{2a}\ln C_{inh} - \frac{1}{2a}\ln K_{ads} \tag{14}$$

where $C$, $\theta$, $a$ and $K_{ads}$ stand for the inhibitor (AMCs) concentration, surface coverage, molecular interaction constant and equilibrium constant, respectively. Based on the values of $K_{ads}$, the standard Gibb's free energy ($\Delta G_{ads}$) values for the adsorption were calculated using the following relationship [20,43]:

$$\Delta G^0_{ads} = -RT\ln(55.5K_{ads}) \tag{15}$$

The calculated values of $K_{ads}$ and $\Delta G_{ads}$ for the adsorption of 5 and 6a calculated at different temperatures are presented in Table 4. Greater $K_{ads}$ values signify a stronger adsorption tendency. According to the findings, AMCs have a strong ability to adsorb on the P110 CS surface in an acidic solution, as indicated by their $K_{ads}$ values, which are in the range of $10^3$. The nature of adsorption has been thoroughly described in the literature using $\Delta G_{ads}$ values. A value of $\Delta G_{ads}$ of $-20$ $kJmol^{-1}$ or higher indicates pure physisorption, while a value of $-40$ $kJmol^{-1}$ or more indicates pure chemisorption. The data reveals that the AMCs' Gads values varied from $-30.19$ to $-33.66$ $kJmol^{-1}$, suggesting their adsorption followed the physiochemisorption mode [20,43].

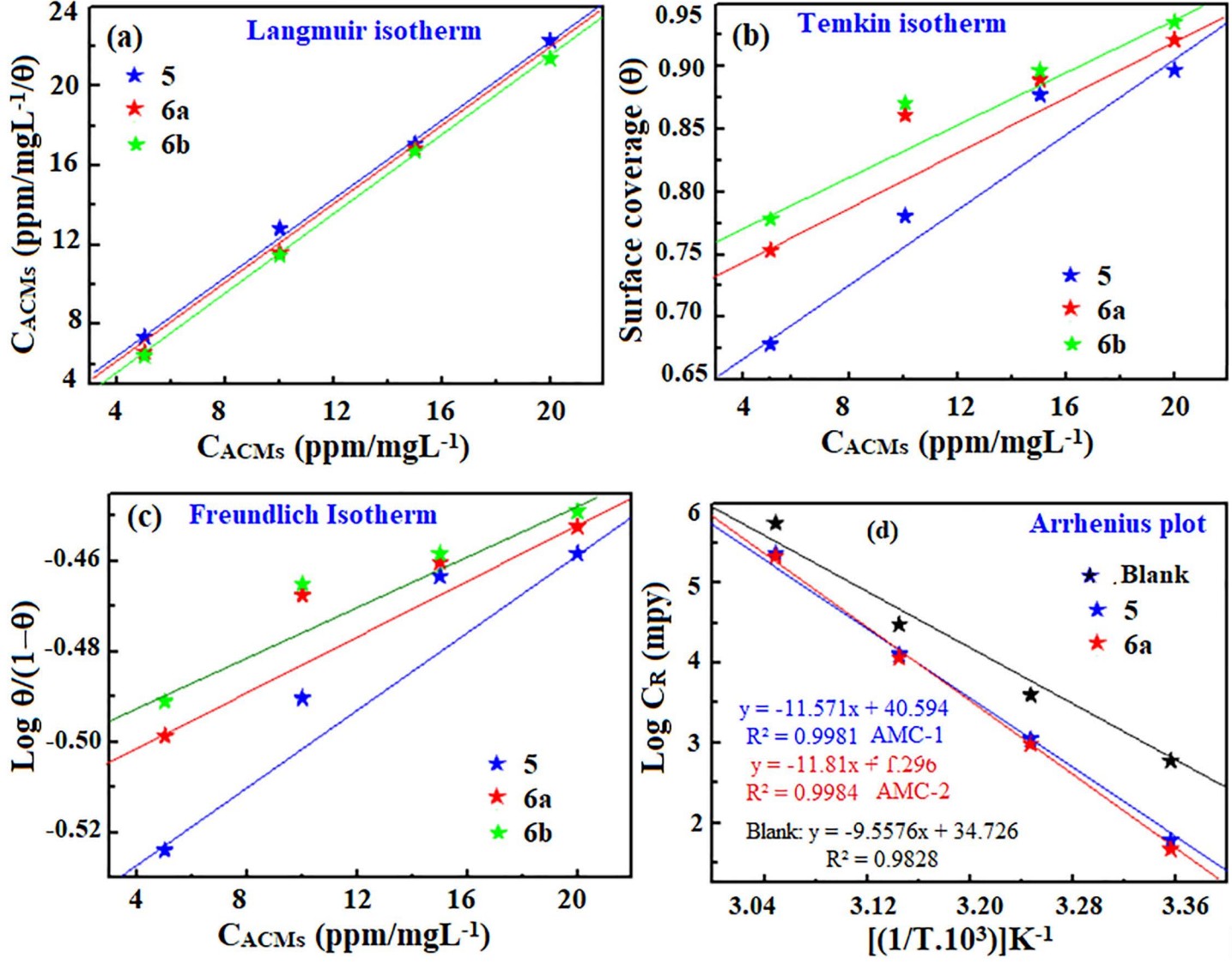

**Fig 5.** (a) Langmuir, (b) Temkin and (c) Freundlich isotherm plots for the adsorption of AMCs on P110 CS surface in 15% HCl. (d) represent the Arrhenius plots for P110 CS corrosion in 15% HCl with and without AMCs (5, 6a and 6b).

The electrochemical studies were also conducted at different temperatures to study the consequences of temperature on the corrosion inhibition potential of AMCs. Rising the temperature can affect the kinetics of the interaction between the corrosive species and the inhibitor molecules in the context of corrosion inhibitors by increasing their mobility [44]. Specific organic inhibitors might perform better at higher temperatures because of better adsorption onto the metal surface, but other organic inhibitors might degrade thermally or have less efficient adsorption [45]. Consequently, a thorough evaluation of the behaviour dependent on temperature is necessary to maximize the effectiveness of organic inhibitors in inhibiting corrosion and creating protective plans for metal substrates under various operational circumstances. The effect of temperature can be best described using the Arrhenius equation [46,47]:

**Table 4. The values of $\log C_R$, surface coverage, $K_{ads}$ and $\triangle G_{ads}$ for the adsorption of AMCs on P110 CS surface in 15% HCl at different temperatures.**

| Temperature | Log corrosion rate (mpy) | | | $\frac{\log \theta}{1} - \theta$ | | $K_{ads}\ (M \times 10^3)$ | | $\triangle G_{ads}$ | |
|---|---|---|---|---|---|---|---|---|---|
| | Blank | 5 | 6a | 5 | 6a | 5 | 6a | 5 | 6a |
| 298 | 2.772 | 1.782 | 1.669 | 0.942 | 1.069 | 10.522 | 14.092 | −32.90 | −33.63 |
| 308 | 3.600 | 3.047 | 2.988 | 0.409 | 0.489 | 3.080 | 3.707 | −30.86 | −31.33 |
| 318 | 4.485 | 4.111 | 4.067 | 0.134 | 0.207 | 1.636 | 1.936 | −30.19 | −30.63 |
| 328 | 5.748 | 5.376 | 5.338 | 0.131 | 0.196 | 1.626 | 1.887 | −31.12 | −31.53 |

$$\log (C_R) = \frac{-E_a}{2.303RT} + \log R \tag{16}$$

The variables $A$, $R$, $E_a$, $R$, $T$, and $C_R$ represent the Arrhenius pre-exponential factor, gas constant, activation energy, absolute temperature, and corrosion rate (mpy), respectively. Fig 5d shows the Arrhenius plot for P110 CS corrosion with and without 5 and 6a. The values of $E_a$ were calculated using the slope values of the plot $(-\frac{E_a}{2.303R})$. The effect of temperature on corrosion rate and surface coverage is presented in Table 4. A rise in temperature leads to a corresponding increase in $C_R$ and a fall in surface coverage. The desorption of adsorbed inhibitor molecules at high temperatures may cause a decrease in protection potential with a temperature increase [48]. Furthermore, the inhibitor molecules may experience chemical alterations, rearrangements, or degradation at high temperatures, which could negatively impact their ability to prevent corrosion [49,50]. More so, because of the higher kinetic energy of inhibitor molecules and the metal surface energy, the intermolecular force of attraction may be reduced at high temperatures. The activation energy values for P110 CS corrosion in 15% HCl solution were $49.54$, $59.97$, and $61.20$ $kJmol^{-1}$ in the absence and 5 and 6a presence, respectively. The higher $E_a$ values in the presence of AMC molecules suggest that the barrier of protection that forms due to their adsorption makes P110 CS corrosion more difficult [51,52].

### 3.4. Surface (SEM and EDX) studies

The effectiveness of corrosion inhibitors on metal surfaces can be evaluated by scanning electron microscopy (SEM) assessment of the surface. SEM offers high-resolution imaging, making it possible to assess the degree of corrosion damage and examine surface morphology. The presence of inhibitor molecules on the metal surface can indicate the creation of protective layers, as shown by the SEM investigation. Corrosion inhibitors work by adhering to metal surfaces and forming a barrier preventing corrosive materials from entering. The existence and location of these inhibitor layers may be visually verified by the SEM examination, which offers important information about the effectiveness of the corrosion prevention mechanism. To fully comprehend the corrosion inhibition process, SEM can also detect any alterations in the microstructure of the metal surface brought about by corrosion and corrosion protection. Fig 6 and SI-11–13 [S1 File] show the SEM images of the polished and corroded P110 CS steel surface with and without the ideal concentrations for a 6-hour immersion time at various magnifications. The polished P110 CS surface has fewer flaws and is generally smoother. There are minor cleaning, polishing, and abrading scratches on the surface. Nevertheless, the strong electrolyte attacks cause severe corrosion and damage to the metallic surfaces upon immersion in a 15% HCl solution. However, it is also evident that AMC molecules, particularly 6b, lead to comparatively reduced corrosion and damage to metallic surfaces. This observation implies that AMCs cover the metal surface to create a protective layer that shields it from corrosive substances and isolates it from aggressive solutions.

The elemental composition of the specimens may be significantly inferred from the energy-dispersive X-ray spectroscopy (EDX) spectrum investigations of corroded and inhibited P110 carbon steel surfaces, both with and without corrosion

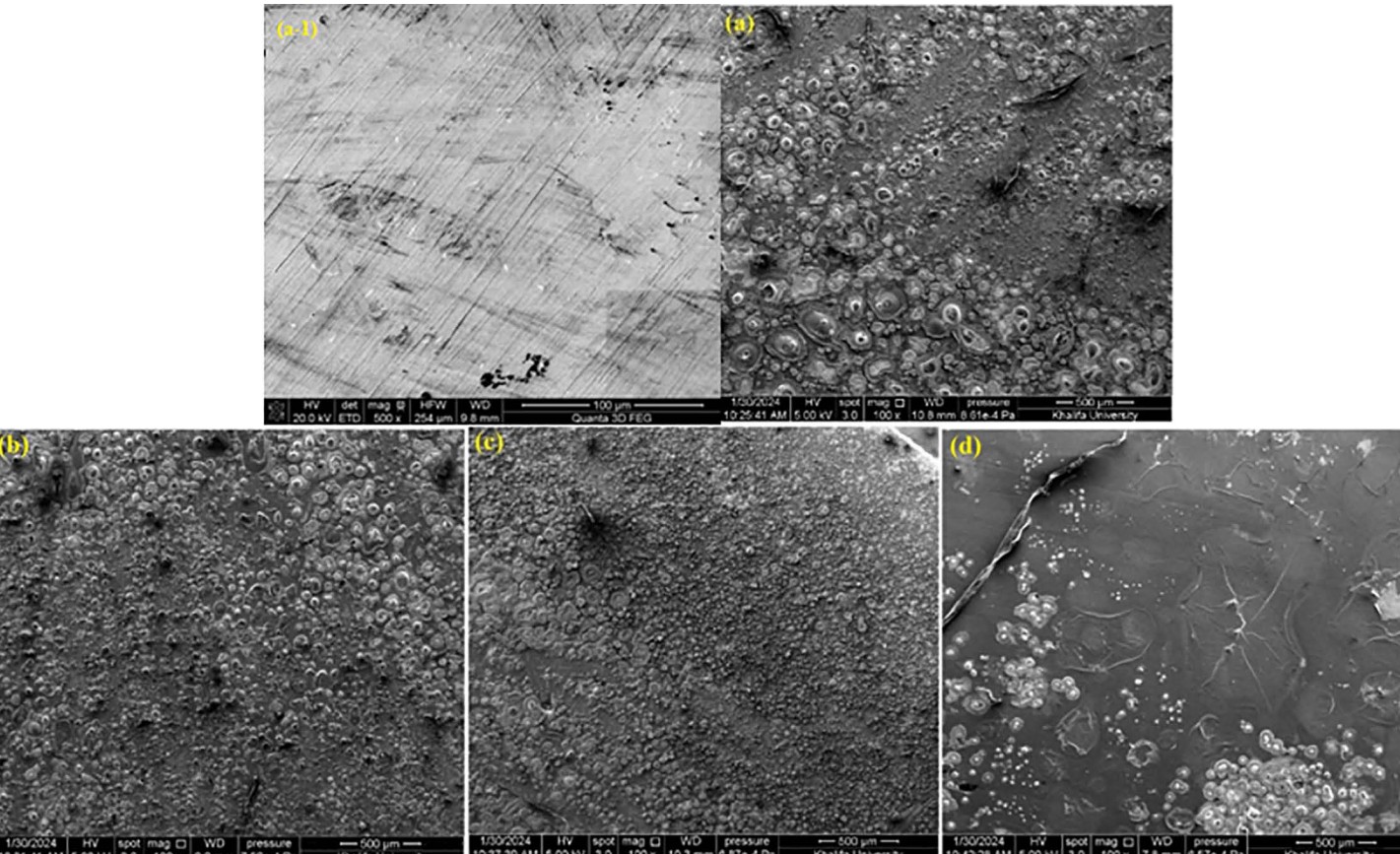

**Fig 6. SEM images (at 100x) of (a-1) polished P110 CS and corroded in 15% HCl for 6h** (a) in the absence and presence of (b) 5, (c) 6a and (d) 6b.

inhibitors. Corroded samples will probably have higher amounts of components in the EDX spectrum that are linked to corrosion byproducts. On the other hand, a distinct elemental proFile should be seen on inhibited surfaces treated with corrosion inhibitors, showing the inhibitors' efficacy in changing the corrosion process. The elements in corrosion inhibitors, metallic elements, and electrolyte elements should be present in the EDX spectra of inhibited samples. It is possible to estimate qualitative information about the adsorption behavior of corrosion inhibitors by comparing the constituents present in shielded and non-protected samples. Fig 7 displays the EDX spectra of polished and corroded surfaces with and without AMCs, and Figure SI-14 [S1 File]. shows the chosen spots based on spectral mapping. Numerous elements, including Fe, Mn, C, P, Cr, and Al, are represented in the EDX spectrum of the polished P110 carbon steel surface. The surface of corroded P110 carbon steel displays an extra chloride signal. The presence of chloride ions in the electrolyte (15% HCl) justifies the existence of chloride. The formation of corrosion products ($Fe_2O_3$ and $Fe_3O_4$) during the extraction of the metallic samples from the electrolyte and their use in EDX tests explains the presence of strong oxygen peaks in all damaged samples. A close examination reveals a distinctive nitrogen peak in every blocked sample. According to this finding, AMC molecules adsorb on the metal surface, forming a layer that prevents corrosion.

### 3.5. DFT study and mechanism of corrosion inhibition

The electronic structure, adsorption energetics, and molecular interactions related to the inhibitory mechanism can be well-understood by DFT investigations [53–55]. To better understand how organic inhibitors, prevent corrosion, DFT

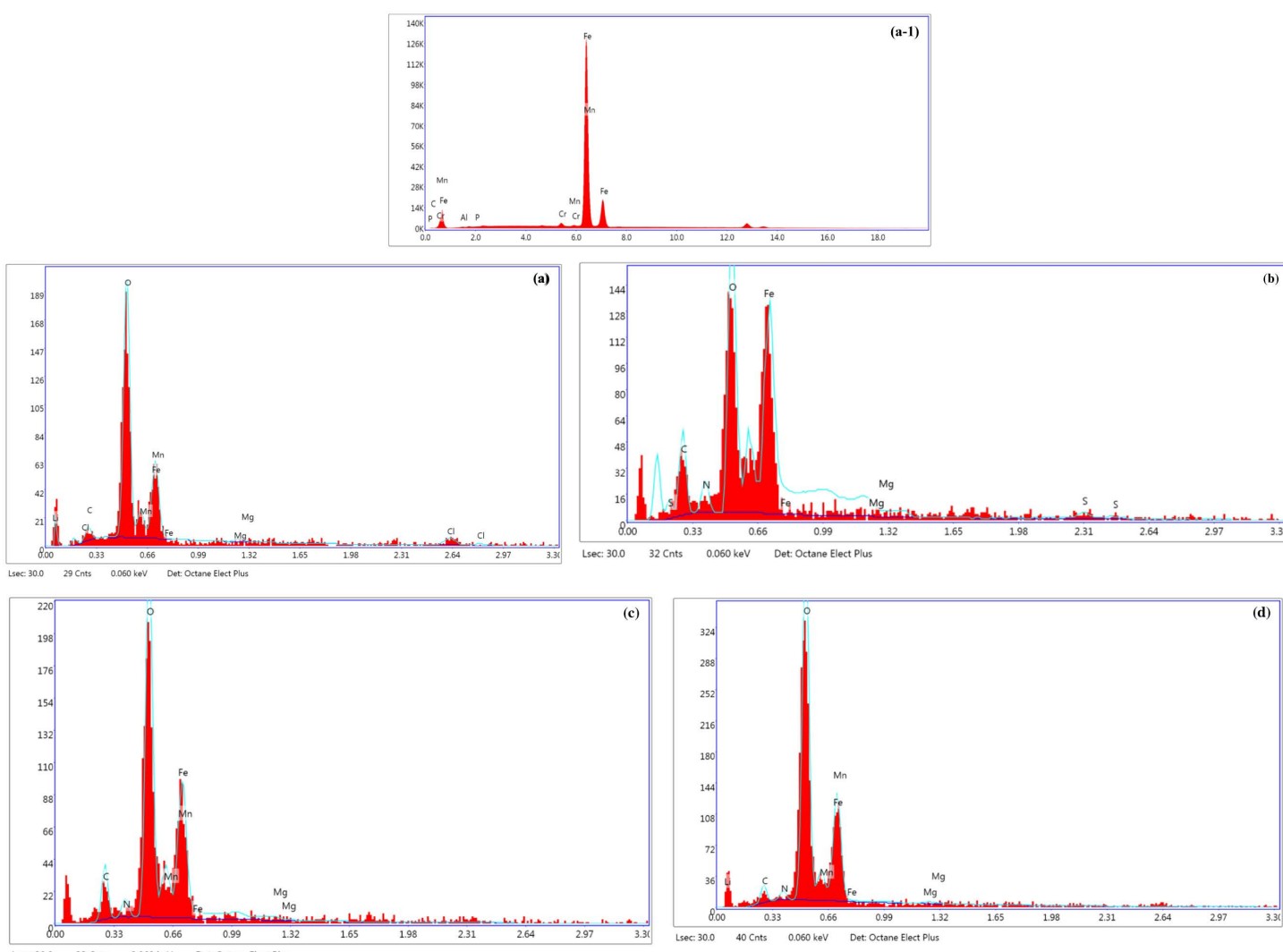

**Fig 7. EDX spectra of (a-1) polished P110 CS and corroded in 15% HCl for 6h** (a) in the absence and presence of (b) 5, (c) 6a and (d) 6b.

makes it possible to accurately forecast important characteristics, including bond formation at the metal-surface bonding, charge transfer, and the energies of frontier molecular orbitals [53–55]. DFT allows for the accurate prediction of critical properties, such as charge transfer, bond formation at the metal-surface contact, and the energies of frontier molecular orbitals, which contributes to our understanding of how organic inhibitors prevent corrosion [53–55]. Two neighboring quaternary ammonium-based rings without (5) and with one (6a) or two (6b) alkyl chains and no alkyl chain $(-(CH_2)_6-^+NMe_2-(CH_2)_{11}-CH_3)$ were used in our current DFT analysis. Their respective assignments are 5, 6a, and 6b. Fig 8 displays their border molecular orbital images. The electron-donating capacity of the inhibitor is represented by the HOMO, signifying that it has the potential to provide electrons to the metal [56,57]. In contrast, the LUMO represents the electron-accepting ability, showing that it may accept electrons from the metal [56,57]. The HOMO-LUMO gap, which measures the energy difference between HOMO and LUMO, affects electron transfer efficiency and provides information about the electronic stability of the inhibitor [56,57]. As may be observed, HOMO/LUMO are primarily found over side chains, quaternary nitrogens of rings, and chloride ions. The finding implies an interaction between these moieties and

 

| Inhibitor | Frontier molecular orbital (FMO) pictures | | |
|---|---|---|---|
| | Optimized | HOMO | LUMO |

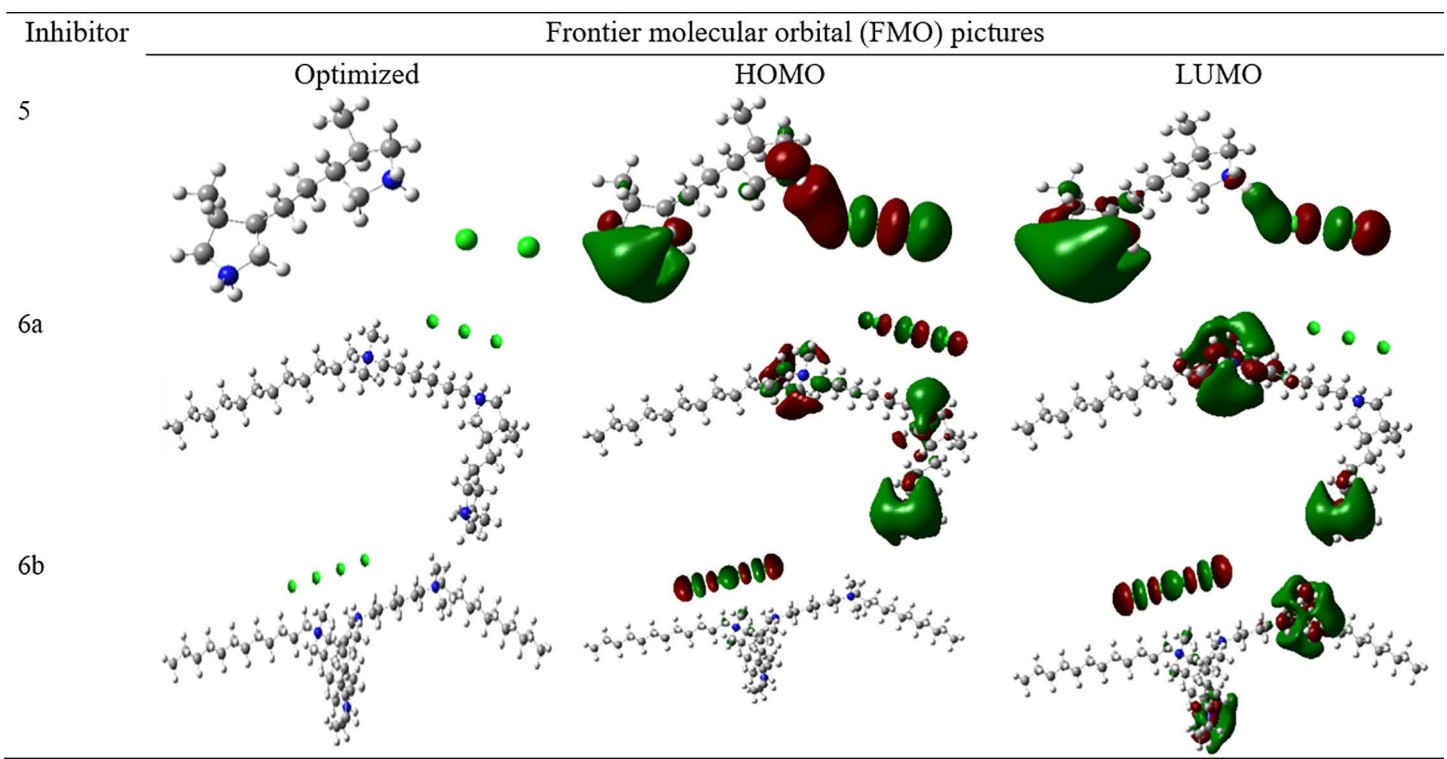

**Fig 8. Frontier molecular orbitals (FMOs; optimized, HOMO, LUMO) pictures of two neighboring quaternary ammonium-based rings without (5) and with one (6a) or two (6b) alkyl chains and no alkyl chain ( $-(CH_2)_6-^+NMe_2-(CH_2)_{11}-CH_3$ ) used in current DFT study.**

the metallic surface. The side chains' quaternary nitrogen atoms are notable for lacking hydrogen, which prevents deprotonation and allows for a sole physisorption method of adsorption on metal surfaces. However, given the right conditions, since quaternary nitrogen atoms in rings have at least one hydrogen atom, they can easily deprotonate, adsorb and bind with the metallic surface through the chemisorption mode [18]. Fig 9 schematically depicts the process of AMC molecules' physisorption and chemisorption. Of course, the hydroxide and chloride ions in the electrolyte can help deprotonate quaternary nitrogen. Table 5 displays the DFT parameters for AMC molecules computed in the solvated phase. A shorter HOMO-LUMO gap often indicates higher electron transfer capability in corrosion prevention, improving adsorption and metal surface protection [21–23]. To forecast and optimize the inhibitory performance of organic compounds and to guide the design of more efficient corrosion inhibitors with specific electronic features, a comprehensive understanding of the HOMO and LUMO characteristics is necessary.

A further examination of the data revealed that 6b has the lowest level of $\Delta E$, which suggests that it has the best inhibitory capability and reactivity [58,59]. In molecular chemistry, hardness is the ability of a molecule to withstand deformation

**Table 5. DFT parameters derived for 5, 6a and 6b in their solvated phase using B3LYP functional and 6-31G (d,p) basis set.**

| $E_{HOMO}$ (eV) | $E_{LUMO}$ (eV) | $\Delta E$ (eV) | IE (eV) | EA (eV) | $\eta$ (eV) | $\chi$ (eV) | $\sigma$ (GPa) | $\Delta N$ | $\mu$ (eV) | Inh |
|---|---|---|---|---|---|---|---|---|---|---|
| −1.2694 | −0.9872 | 0.2821 | 1.2694 | 0.9872 | 0.1410 | 1.1283 | 7.0876 | 13.0826 | 37.3198 | 5 |
| −0.3942 | −0.1483 | 0.2459 | 0.3942 | 0.1483 | 0.1229 | 0.2712 | 8.1303 | 18.4913 | 50.5826 | 6a |
| −0.6095 | −0.3986 | 0.2108 | 0.6095 | 0.3986 | 0.1054 | 0.5040 | 9.4836 | 20.4653 | 89.7425 | 6b |

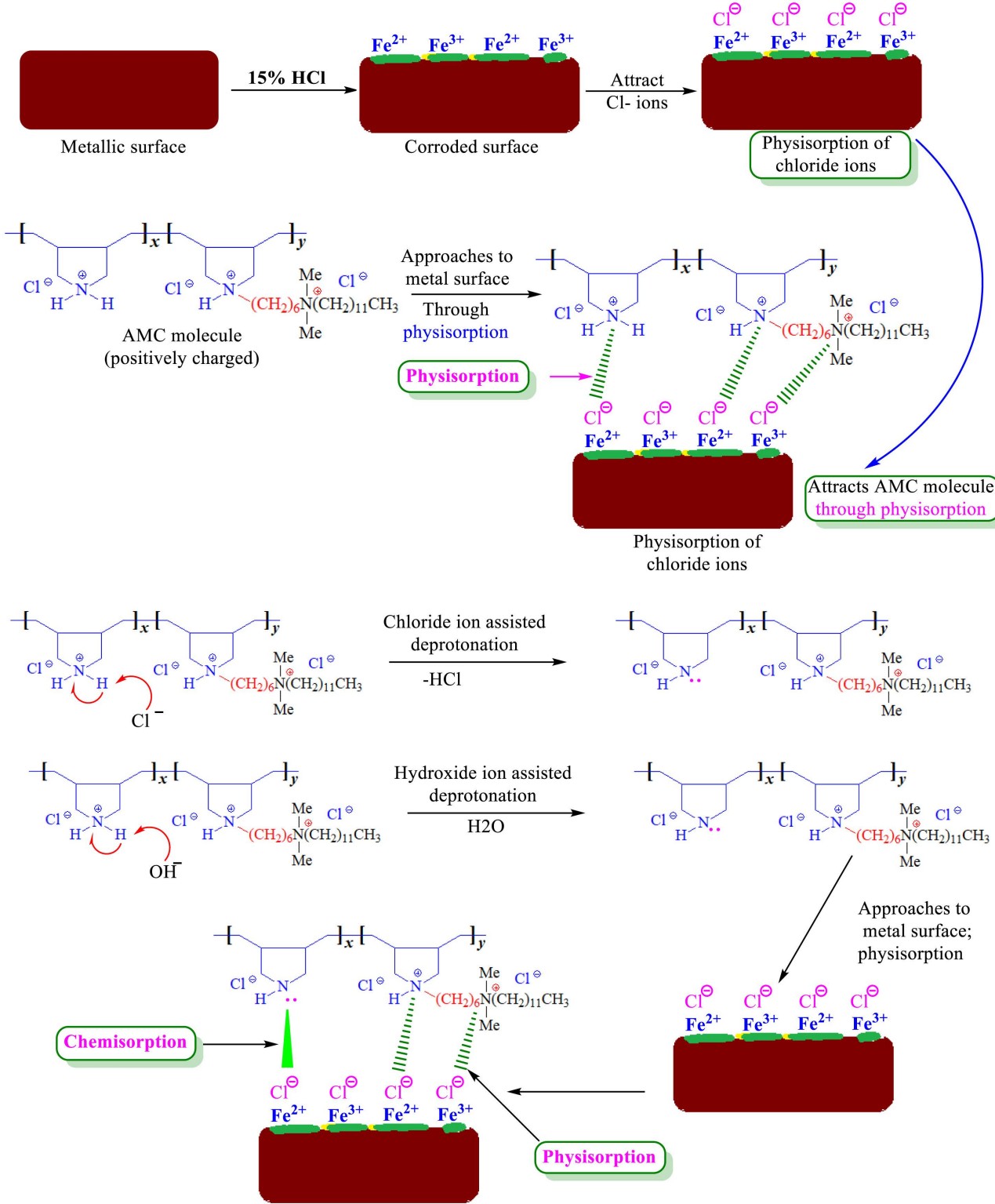

**Fig 9. Schematic illustration of physical and chemical adsorption of AMC molecules on P110 CS surface and the formation of corrosion inhibitive film.**

or electron redistribution. A hard molecule is less likely to donate or take electrons and has more stable orbitals. Since hardness and softness affect adsorption intensity and the creation of a barrier of protection on the metal, achieving a balance between them is essential for effective inhibition. The corrosion inhibition potential trend reported in the current investigation agrees with the hardness and softness values [60,61]. It is also anticipated that a molecule with less electronegativity may work better as a corrosion inhibitor, coordinating with a metallic surface. Our latest research has shown no consistent pattern in the electronegativity levels. Stronger electrostatic forces with metal surfaces result from a more significant dipole moment, which signifies greater charge separation within the molecule. The organic inhibitor shows improved adsorption onto the metal with increased dipole moment, forming a more robust protective layer [62,63]. The dipole moment values justified the experimental order of inhibitory efficiency of AMC molecules. The ability of the inhibitor to donate or accept electrons throughout adsorption on the metal surface is indicated by a more significant electron transfer fraction [21–23]. Corrosion rates are decreased by this electron transfer, which promotes the development of a persistent protective layer. Additionally, the measurements of the fraction of electron transfer support the experimental order of inhibition potential.

## 4. Conclusions

The present study describes the inhibition potential of three quaternary ammonium-based copolymers (AMCs) having varying degrees of hydrophobicity for P110 CS corrosion in 15% HCl. The following conclusions were drawn:

1. AMC molecules act as effective corrosion inhibition, and their effectiveness increases with increasing their concentrations.

2. Changes in their hydrophobic properties significantly impact the adsorption and inhibition effectiveness, corrosion rate and surface coverage values of AMC molecules.

3. The results showed that AMCs with hydrophilic and hydrophobic ratios of 100 (5), 90:10 (6a), and 80:20 (6b) showed the %*IE of* $87.74\%$, $92.12\%$, *and* $93.53\%$, respectively.

4. The OCP and PDP studies demonstrate that AMCs effectively replace the pre-adsorbed water molecules at the active areas of the metallic surface.

5. AMC molecules retard both anodic and reactions and behave as mixed-type corrosion inhibitors.

6. They work by forming a corrosion-inhibiting coating on the metal surface through adsorption, based on the Langmuir isotherm concept.

7. Surface examinations such as SEM and EDX studies showed that AMCs build a protective film at the metal and electrolyte interface.

8. DFT analysis indicates that the adsorption and charge-sharing processes involve a significant involvement from quaternary nitrogen atoms of hydrophilic and hydrophobic moieties.

## Supporting information

**S1 File.** **Figure SI-1.** FTIR spectrum of quaternary ammonium bromide **3**. **Figure SI-2.** (a) [1]H and (b) [13]C NMR spectra of quaternary ammonium bromide **3** in $D_2O$. **Figure SI-3.** FTIR spectrum of homopolymer **5**. **Figure SI-4.** [1]H NMR spectra of (a) monomer **4** and (b) homopolymer **5** in $D_2O$. **Figure SI-5.** [13]C NMR spectra of (a) monomer **4** and (b) homopolymer **5** in $D_2O$. **Figure SI-6.** FTIR spectrum of copolymer **6b**. **Figure SI-7.** FTIR spectrum of copolymer **6b**. **Figure SI-8.** [1]H NMR spectra of (a) monomer **4** in $D_2O$, and (b) **6b** in $CD_3OD$. **Figure SI-9.** [13]C NMR spectra of (a) monomer **4** in $D_2O$, and (b) **6b** in $CD_3OD$. **Figure SI-10.** TGA curves of homopolymer **5** and copolymers **6a** and **6b**. **Figure SI-11**: SEM images (at

500x) of corroded P110 CS in 15% HCl for 6h (a) in the absence and presence of (b) **5** (c) **6a** and (d) **6b**. **Figure SI-12**: SEM images (at 1000x) of corroded P110 CS in 15% HCl for 6h (a) in the absence and presence of (b) **5**, (c) **6a** and (d) **6b**. **Figure SI-13**: SEM images (at 2000x) of corroded P110 CS in 15% HCl for 6h (a) in the absence and presence of (b) **5**, (c) **6a** and (d) **6b**. **Figure SI-14**: Pictures showing sites taken for EDX analyses from corroded P110 CS samples (a) in the absence and presence of (b) **5**, (c) **6a** and (d) **6b**.
(DOCX)

## Author contributions

**Conceptualization:** Imad Barsoum, Akram Alfantazi.

**Data curation:** Mohammad A. Jafar Mazumder.

**Investigation:** Chandrabhan Verma, Ghadeer Mubarak.

**Methodology:** Chandrabhan Verma, Ghadeer Mubarak, Mohammad A. Jafar Mazumder.

**Resources:** Ghadeer Mubarak.

**Software:** Ghadeer Mubarak.

**Supervision:** Imad Barsoum, Akram Alfantazi.

**Validation:** Mohammad A. Jafar Mazumder.

**Visualization:** Akram Alfantazi.

**Writing – original draft:** Chandrabhan Verma.

**Writing – review & editing:** Imad Barsoum, Akram Alfantazi.

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
