## [Decision Letter · Decision Letter 0]

Dear Dr. Verma,

Thank you for submitting your manuscript to PLOS ONE. After careful consideration, we feel that it has merit but does not fully meet PLOS ONE’s publication criteria as it currently stands. Therefore, we invite you to submit a revised version of the manuscript that addresses the points raised during the review process.

We look forward to receiving your revised manuscript.

Kind regards,

Prashant Singh

Academic Editor

PLOS ONE

Journal Requirements:

Additional Editor Comments :

Reviewers have submitted their opinion on manuscript and asked to revise the manuscript.

Reviewers' comments:

Reviewer's Responses to Questions

**Comments to the Author**

1. Is the manuscript technically sound, and do the data support the conclusions?

Reviewer #1: Yes

Reviewer #2: Yes

Reviewer #3: Yes

2. Has the statistical analysis been performed appropriately and rigorously?

Reviewer #1: Yes

Reviewer #2: Yes

Reviewer #3: N/A

3. Have the authors made all data underlying the findings in their manuscript fully available?

Reviewer #1: Yes

Reviewer #2: Yes

Reviewer #3: Yes

4. Is the manuscript presented in an intelligible fashion and written in standard English?

Reviewer #1: Yes

Reviewer #2: Yes

Reviewer #3: Yes

Reviewer #1: Mubarak et al. synthesized three quaternary ammonium-based copolymers (AMCs) with different hydrophobic qualities to inhibit P110 CS corrosion in 15% HCl. The results showed that AMCs act as corrosion inhibitors, with over 90% inhibition efficiency at 20 ppm concentration. They act as mixed-type corrosion inhibitors. The work is interesting and should be considered for publication after addressing the following comments:

(a) The study's objectives should be further explored in the introduction.

(b) The author should focus on how P110 CS is preferred over other steel grades for this application.

(c) In the introduction, I recommend adding the corrosion mechanism in 15% HCl. This would be highly useful.

(d) Please describe the reproducibility of experimental data.

(e) The manuscript should be checked for grammatical and syntax error corrections.

(f) Check and correctly write the units in Tables 1 and 2.

After addressing the above minor comments, the manuscript can be accepted without further review.

Reviewer #2: The present study describes the corrosion inhibition effect of three quaternary ammonium-based copolymers (AMCs) for P110 CS in 15% HCl. The study reported the effect of hydrophobic character on their inhibition effect. The inhibition potential increases with the hydrophobic character. The inhibitors are highly effective, showing over 90% efficiency at 20 ppm. My review comments and suggestions are given below:

1.    Generally, such large molecules face solubility problems in polar electrolytes. What was their solubility? Add a few lines to it in the experimental section.

2.    Please highlight the selection criteria for electrolytes, electrolytes, and inhibitors.

3.    Please enlarge the discussion on the hydrophobic/hydrophilic ratio on corrosion and corrosion protection.

4.    The inhibitors are positively charged/cationic. Please justify how they are adsorbing using chemisorption mode and following the Langmuir isotherm.

5.    The corrosion protection mechanism is described in detail; however, no information has been given on the transport phenomenon of inhibitors to the metal surface.

6.    Can activation energy's value be linked with the inhibitors' adsorption mode? Please read and cite some recent literature.

7.    Report the standard deviation in tables for parameters derived experimentally.

Reviewer #3: The authors presented an intelligible findings that provide insights for the or casing and

tubing using three identify and formulate corrosion

inhibitors namely

quaternary ammonium-based copolymers (AMCs) with different hydrophobic qualities and

investigate their ability to inhibit P110 Carbon Steel corrosion in 15% HCl environments.

**Do you want your identity to be public for this peer review?** For information about this choice, including consent withdrawal, please see our Privacy Policy

Reviewer #1: **Yes: ** Valentine Chikaodili Anadebe

Reviewer #2: No

Reviewer #3: No

---

## [Author Response · Author response to Decision Letter 1]

17 Feb 2025

Reviewer #1:

Reviewer’s Comments: Mubarak et al. synthesized three quaternary ammonium-based copolymers (AMCs) with different hydrophobic qualities to inhibit P110 CS corrosion in 15% HCl. The results showed that AMCs act as corrosion inhibitors, with over 90% inhibition efficiency at 20 ppm concentration. They act as mixed-type corrosion inhibitors. The work is interesting and should be considered for publication after addressing the following comments:

Author’s Response: Dear reviewer, thank you for reviewing our manuscript and making valuable comments and suggestions. The comments and suggestions were very constructive, and they have greatly improved the quality of the revised manuscript. We revised our manuscript based on your reviewers' comments and suggestions; the revised areas are highlighted in red. In the response letter, we reproduced all comments by heading “Reviewer’s Comment”; our responses are given by heading “Author’s Response”. We believe that the revised manuscript meets the editor’s expectations and the journal’s standards.

Reviewer’s Comment (a): The study's objectives should be further explored in the introduction.

Author’s Response: Dear reviewer, thank you for your valuable comments. The study's objectives are highlighted further in the last paragraph of the introduction.

Reviewer’s Comment (b): The author should focus on how P110 CS is preferred over other steel grades for this application.

Author’s Response: Dear reviewer, thank you for your useful suggestion. The introduction has been revised, and more information on the preferential use of P110 CS for casing and tubing applications has been added.

Reviewer’s Comment (c): In the introduction, I recommend adding the corrosion mechanism in 15% HCl. This would be highly useful.

Author’s Response: Dear reviewer, we appreciate your comments. However, the mechanism of steel corrosion in acidic solutions is frequently reported. Therefore, we don’t think there would be any benefits to adding a mechanism.

Reviewer’s Comment (d): Please describe the reproducibility of experimental data.

Author’s Response: Dear reviewer, the electrochemical studies were repeated many times until we got consistent readings.

Reviewer’s Comment (e): The manuscript should be checked for grammatical and syntax error corrections.

Author’s Response: Dear reviewer, thank you for your suggestion. The entire manuscript has been checked carefully, and grammatical and syntax errors have been corrected.

Reviewer’s Comment (f): Check and correctly write the units in Tables 1 and 2. After addressing the above minor comments, the manuscript can be accepted without further review.

Author’s Response: Dear reviewer, thank you for your suggestion. The units have been checked and corrected in Tables 1 and 2.

Reviewer #2:

Reviewer’s Comments: The present study describes the corrosion inhibition effect of three quaternary ammonium-based copolymers (AMCs) for P110 CS in 15% HCl. The study reported the effect of hydrophobic character on their inhibition effect. The inhibition potential increases with the hydrophobic character. The inhibitors are highly effective, showing over 90% efficiency at 20 ppm. My review comments and suggestions are given below:

Author’s Response: Dear reviewer, thank you for reviewing our manuscript and making valuable comments and suggestions. The comments and suggestions were very constructive, and they have greatly improved the quality of the revised manuscript. We revised our manuscript based on your reviewers' comments and suggestions; the revised areas are highlighted in red. In the response letter, we reproduced all comments by heading “Reviewer’s Comment”; our responses are given by heading “Author’s Response”. We believe that the revised manuscript meets the editor’s expectations and the journal’s standards.

Reviewer’s Comment 1: Generally, such large molecules face solubility problems in polar electrolytes. What was their solubility? Add a few lines to it in the experimental section.

Author’s Response: Dear reviewer, thank you for your useful suggestion. Because of their cationic nature, the evaluated inhibitors were completely soluble in the test electrolyte within the investigated concentration range. This information has been added to the revised manuscript. Please see heading 2.2.

Reviewer’s Comment 2: Please highlight the selection criteria for electrolytes, electrolytes, and inhibitors.

Author’s Response: Dear reviewer, thank you for your useful suggestion. The criteria behind the selection of electrolytes, electrolytes, and inhibitors have been explored in the introduction.

Reviewer’s Comment 3: Please enlarge the discussion on the hydrophobic/hydrophilic ratio on corrosion and corrosion protection.

Author’s Response: Dear reviewer, thank you for your useful suggestion. The discussion on hydrophobic/hydrophilic ratio on corrosion and corrosion protection.

Reviewer’s Comment 4: The inhibitors are positively charged/cationic. Please justify how they are adsorbing using chemisorption mode and following the Langmuir isotherm.

Author’s Response: Dear reviewer, thank you for your useful comment and suggestion. You are kindly requested to read the mechanism of the corrosion inhibition part. There, we have described that the protonated inhibitor molecules approach the metal surface through physisorption, but once they reach the metal surface, the unshared electron pairs may be transferred via coordination bonding or chemisorption mode.

Reviewer’s Comment 5: The corrosion protection mechanism is described in detail; however, no information has been given on the transport phenomenon of inhibitors to the metal surface.

Author’s Response: Dear reviewer, we appreciate your comments. However, the mechanism of steel corrosion in acidic solutions is frequently reported. Therefore, we don’t think there would be any benefits to adding a mechanism.

Reviewer’s Comment 6: Can activation energy's value be linked with the inhibitors' adsorption mode? Please read and cite some recent literature.

Author’s Response: Dear reviewer, the significance of activation energy values with adsorption behavior has been described in the last segment of heading 3.3.

Reviewer’s Comment 7: Report the standard deviation in tables for parameters derived experimentally.

Author’s Response: Dear reviewer, the electrochemical studies were repeated many times until we got consistent readings. The goodness of fit values have been given in the tables.

Reviewer #3:

Reviewer’s Comments: The authors presented intelligible findings that provide insights for the or casing and

tubing using three identify and formulate corrosion inhibitors namely

quaternary ammonium-based copolymers (AMCs) with different hydrophobic qualities and

investigate their ability to inhibit P110 Carbon Steel corrosion in 15% HCl environments.

Author’s Response: Author’s Response: Dear reviewer, thank you so much for reviewing our manuscript and recommending it for consideration.

---

## [Decision Letter · Decision Letter 1]

Exploring the Hydrophobic Effects of Quaternary Ammonium Copolymers on Corrosion of Casing and Tubing Steel in Acidic Solution

PONE-D-25-05283R1

Dear Dr. Verma,

We’re pleased to inform you that your manuscript has been judged scientifically suitable for publication and will be formally accepted for publication once it meets all outstanding technical requirements.

Kind regards,

Prashant Singh

Academic Editor

PLOS ONE

Additional Editor Comments (optional):

Accept

Reviewers' comments:

Reviewer's Responses to Questions

**Comments to the Author**

Reviewer #1: All comments have been addressed

Reviewer #2: All comments have been addressed

2. Is the manuscript technically sound, and do the data support the conclusions?

Reviewer #1: Yes

Reviewer #2: Yes

3. Has the statistical analysis been performed appropriately and rigorously?

Reviewer #1: Yes

Reviewer #2: Yes

4. Have the authors made all data underlying the findings in their manuscript fully available?

Reviewer #1: Yes

Reviewer #2: Yes

5. Is the manuscript presented in an intelligible fashion and written in standard English?

Reviewer #1: Yes

Reviewer #2: Yes

Reviewer #1: Author have significantly improved on the revised version of this manuscript. I strongly recommend acceptance in its current form.

Reviewer #2: Authors have addressed all the comments made by reviewer in the revised manuscript and now acceptable for publication,

**Do you want your identity to be public for this peer review?** For information about this choice, including consent withdrawal, please see our Privacy Policy

Reviewer #1: **Yes: ** Valentine Chikaodili Anadebe

Reviewer #2: **Yes: ** Indra Bahadur

---

## [Editor Report · Acceptance letter]

PONE-D-25-05283R1

PLOS ONE

Dear Dr. Verma,

I'm pleased to inform you that your manuscript has been deemed suitable for publication in PLOS ONE. Congratulations! Your manuscript is now being handed over to our production team.

Kind regards,

on behalf of

Dr. Prashant Singh

Academic Editor

PLOS ONE